**METHODS AND APPROACHES**

# Artificial intelligence approaches to the volumetric quantification of glycogen granules in EM images of human tissue

Eduardo Ríos[1], Montserrat Samsó[2], Lourdes C. Figueroa[1], Carlo Manno[1], Eshwar R. Tammineni[1], Lucas Rios Giordano[3], and Sheila Riazi[4]

**Skeletal muscle, the major processor of dietary glucose, stores it in myriad glycogen granules. Their numbers vary with cellular location and physiological and pathophysiological states. AI models were developed to derive granular glycogen content from electron-microscopic images of human muscle. Two UNet-type semantic segmentation models were built: "Locations" classified pixels as belonging to different regions in the cell; "Granules" identified pixels within granules. From their joint output, a pixel fraction *pf* was calculated for images from patients positive (MHS) or negative (MHN) to a test for malignant hyperthermia susceptibility. *pf* was used to derive *vf*, the volume fraction occupied by granules. The relationship *vf* (*pf*) was derived from a simulation of volumes ("baskets") containing virtual granules at realistic concentrations. The simulated granules had diameters matching the real ones, which were measured by adapting a utility devised for calcium sparks. Applying this relationship to the *pf* measured in images, *vf* was calculated for every region and patient, and from them a glycogen concentration. The intermyofibrillar spaces and the sarcomeric I band had the highest granular content. The measured glycogen concentration was low enough to allow for a substantial presence of non-granular glycogen. The MHS samples had an approximately threefold lower concentration (significant in a hierarchical test), consistent with earlier evidence of diminished glucose processing in MHS. The AI models and the approach to infer three-dimensional magnitudes from two-dimensional images should be adaptable to other tasks on a variety of images from patients and animal models and different disease conditions.**

## Introduction

The present article describes methods that use artificial intelligence (AI) to automate the task of locating glycogen granules for the ultimate goal of quantifying glycogen content in electron microscopic (EM) images of skeletal muscle cells. Two different approaches to image analysis were initially used: categorical classification (CC) (Byerly et al., 2022) and semantic segmentation (SS) (Csurka et al., 2023, *Preprint*). While both types of models achieved the tasks at hand, the categorical models met substantial limitations. The segmentation models, by contrast, had none of those limitations, and constitute the main output of this methodological study.

While the methods that will be presented were designed for the specific quantification of glycogen granules, they should be adaptable to the automation and objective accomplishment of other tasks, including the computation of frequency and density of a number of structures (static) or events (kinetic) that can be imaged, electron microscopically or otherwise. Any of the methods to be described, if applied with different targets for analysis, will of course require additional efforts in the

adaptation of network structures and their training. The present work aims to provide a practical starting platform by including proven network structures in pretrained form, code presented in supplements, and optimized parameters ("weights") deposited in a public database.

An additional feature of note in this study, prompted by observations from the journal reviewers, is a solution to the problem of inferring granular content and glycogen concentrations—three dimensional magnitudes—from EM images—flat, two dimensional objects. The approach, signaled by the word "volumetric" in the title, should be applicable to other transitions from images to 3-D variables when a more direct approach (e.g., *z*-stacks) is not available.

### The physiological question

Skeletal muscle is the major processing site for dietary glucose, which muscle consumes at high rates during exercise and stores as glycogen, more so than any other tissue (Baron et al., 1988; Mizgier et al., 2014). These functions place skeletal muscle as a

---

[1]Department of Physiology and Biophysics, Rush University, Chicago, IL, USA;   [2]Department of Physiology and Biophysics, Virginia Commonwealth University, Richmond, VA, USA;   [3]LRG Architects, Santa Monica, CA, USA;   [4]Department of Anesthesia and Pain Management, University of Toronto, Toronto, ON, Canada.

Correspondence to Eduardo Rios: erios@rush.edu.



critical agent for glucose homeostasis, a role that many deem equally important as the generation of force and movement (e.g., Ozawa, 2011). The relevance of this metabolic role is reflected in the fact that failure of muscle intake of glucose constitutes the main proximate cause of hyperglycemia and type-2 diabetes (DeFronzo and Tripathy, 2009).

Muscle glycogen content, 20–30 g per kilogram of dry weight in sedentary humans (Burke et al., 2017; Marchand et al., 2007), is highly positively correlated with exercise performance. While the reason for the close relationship is not fully understood, it is known that both the contractile and ion transport functions of muscle are directly dependent on ATP, and that maintenance of ATP levels when it is being consumed rapidly depends critically on the availability of glycogen. As reviewed by Ørtenblad and Nielsen (2015), muscle function depends critically on the adequate concentrations of glycogen at various cellular locations. The underlying concept is that energy metabolism is compartmentalized in skeletal myocytes; for glycogen to supply the ATP needed locally, it must be near the enzymes that allow its synthesis from, and lysis to, glucose, and close to the cellular machinery that utilizes ATP. Two aspects of glycogen distribution reflect these constraints: glycogen is largely present in roughly spherical granules that include enzymes and regulatory proteins in their composition (reviewed by Prats et al., 2018); also, the granules are distributed at different densities in different regions of the cell.

The heterogeneous distribution of glycogen granules poses multiple questions, most of which have been addressed for decades in the vast literature of the field. It has been shown most recently by Nielsen et al. (2022) that glycogen in granules at different locations is used preferentially by different ATPases. Therefore, the observations suggest that glycogen at different locations serves different functions. However, they seem to be more abundant in the I band than the A band, a preference that does not have an obvious functional justification. Granules may appear at various locations, often in enormously large numbers. A fast, automated identification and quantification of granules, together with their automatic regional location, will allow an objective comparison of concentrations. After AI models are adequately trained, they allow rapid and effortless analysis of thousands of images. Given their potential to quantify millions of granules, these procedures will afford comparisons—between regions or between different metabolic or pathologic conditions—of a precision that cannot be reached by the existing manual methods.

While most research on glycogen has focused on granules, there is evidence of glycogen in other forms and locations. The evidence, however, is largely indirect, including the association of glycogen with various proteins in the SR and myofilaments (e.g., Cuenda et al., 1994, 1995; Caudwell et al., 1978). More recently, Tammineni et al. (2020) demonstrated that glycogen phosphorylase (GP), the rate-limiting enzyme for glycogen utilization, is distributed over the sarcoplasmic reticulum (SR) membranes as a homogeneous-looking layer, again suggesting the presence of a distributed form of glycogen. There is, however, no direct demonstration or quantification of this form. The uncertainty could be removed by a comparison between total

glycogen content, measured by chemical analysis, and the content in granules. To this end, a precise and accurate quantification of glycogen in granules under different treatments would be required. The automatic procedure developed here constitutes a step toward this goal.

**The pathophysiological question**
The method can be applied to characterize features associated with disease. Here, it is applied to images of muscle biopsies from patients subjected to the caffeine-halothane contracture test (CHCT) diagnostic for susceptibility to malignant hyperthermia (MHS). MHS (Litman and Rosenberg, 2005; Litman et al., 2018) is thought to be caused by a primary "$Ca^{2+}$ leak" from the SR, which in turn leads to a steady increase in $[Ca^{2+}]_{cytosol}$ (Lopez et al., 1986, 1992). Among the consequences of long-term increase in $[Ca^{2+}]_{cytosol}$, MHS muscles show changes in the activity of glycogen synthesis and lysis enzymes, leading to a shift in the glycogen < - > glucose balance. The quantification of glycogen granules in these patients constitutes an ideal field for the application of the present method. Many other diseases course with different degrees of such "calcium stress" in both skeletal (Edwards et al., 2010; Boittin et al., 2006; Bellinger et al., 2009) and cardiac muscle (Guo et al., 2018; Lahiri et al., 2020) and are therefore candidates for disruption of glucose/glycogen balance, quantifiable by the present analysis.

## The data
The data analyzed consists of a number of transmission EM images of ultrathin sections of biopsied gracilis muscle taken from individuals subjected to CHCT in the Malignant Hyperthermia Investigation Unit (MHIU) of the Canadian University Health Network (Toronto). Sections were fixed slack or slightly stretched and stained by a standard heavy metals technique (see Materials and methods) to which a step of treatment with potassium ferrocyanide ($K_4Fe(CN)_6$) was added to visualize glycogen. Images were acquired at various magnifications (ranging from 0.4 to 1.4 nm per pixel), in three different microscopes in three laboratories, as arrays of 4,096 × 4,096 or 2,048 × 2,048 pixels. This range of magnifications was adequate to discern glycogen granules within their surrounding subcellular context, consisting of myofilaments, SR, T-tubules, and mitochondria within the repetitive sarcomeric structure of skeletal muscle. A representative image stained for glycogen is shown in Fig. 1 A. Part of a section prepared similarly but omitting the specific staining is shown in Fig. 1 B. Fig. 1 A shows abundant glycogen in roughly spherical granules of diameter between 15 and 40 nm. Clear areas, roughly circular and of diameters in the same range, appear in the unstained images. Focusing on the I bands, where the glycogen granules appear aligned in the axial direction, the similar arrangement of clear areas indicates that they correspond to unstained glycogen granules (Fig. 1 B). Also notable in the ferrocyanide-stained images is the intensity of staining of SR (white arrow) and transverse tubular (TT) membranes (black arrow), a feature not present in our unstained images, but variably found in previous work, e.g., Eisenberg et al. (1974),

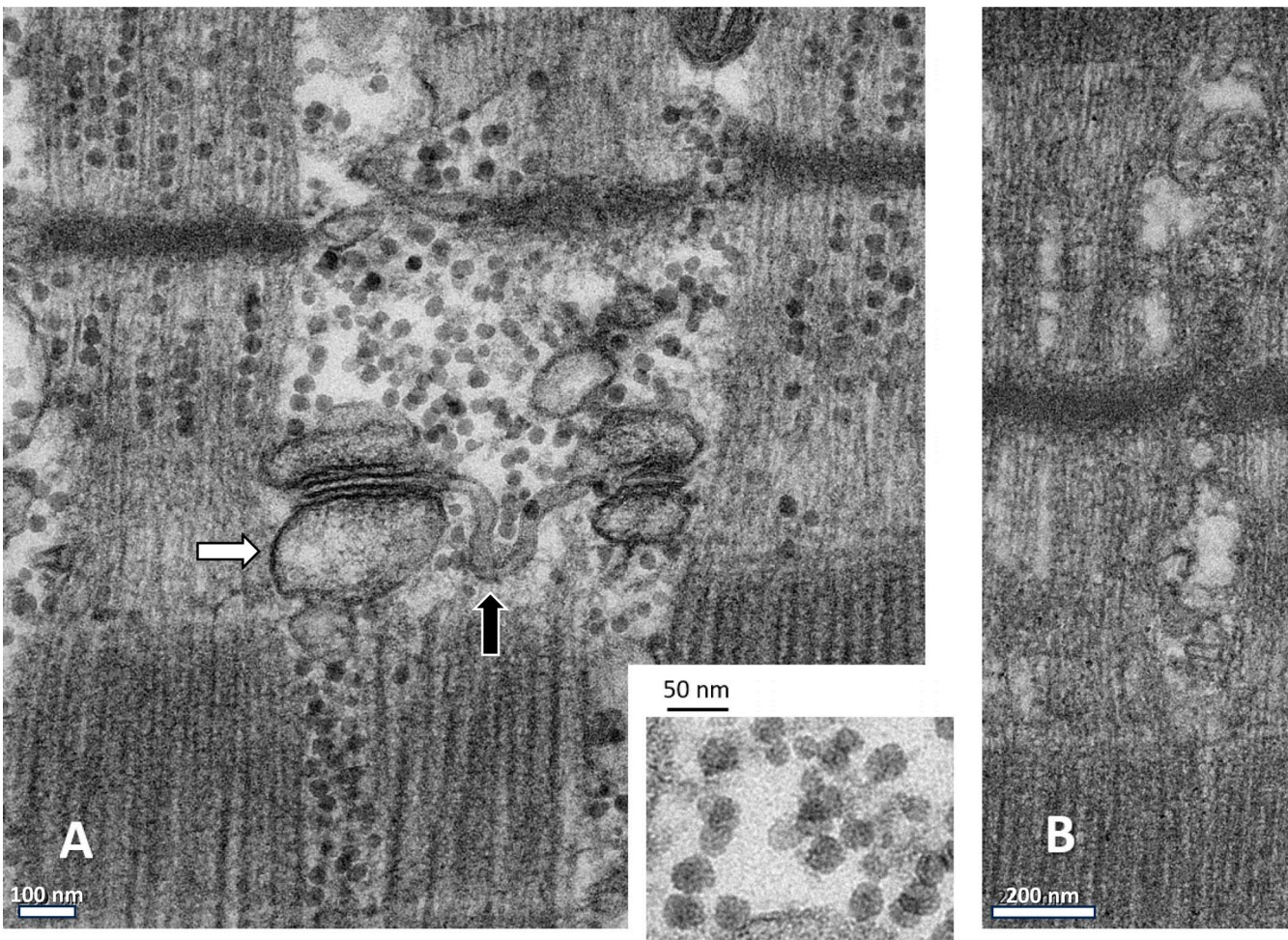

Figure 1.   **Example EM images. (A and B)** A comparison of images of ultrathin sections stained for glycogen (A) or not (B). The stained image shows granules of 15–40 nm (enlarged in inset). The corresponding unstained image, of a section cut from the same block, shows roughly circular clear areas separating thin myofilaments, consistent with unstained granules. The image in A also shows a thick deposition of staining on SR and junctional T tubule membranes, not present in B.

suggesting the presence of a layer of glycogen, especially at or near junctional regions.

The tissues analyzed belong to seven patients who consented to the study, subjected in the clinic to the CHCT; their identities are only known to the clinic personnel, and their demographic and clinical data are listed in Table 1. The sole selection criterion for this initial study was the availability of tissue; consequently, their diagnosis status was mixed. Four of the seven patients had a positive diagnosis of susceptibility to malignant hyperthermia (MHS) and the other three had normal reactions (MHN). A "clinical index" that combines subjective symptoms and objective analyses (Figueroa et al., 2019) reached low levels (0 or 1.67 in a 0–10 scale) for all the MHN and two of the MHS patients, and a high value (6.67—on account of complaints for serious cramps, heat stroke, and rhabdomyolysis) for patients # 143 and 145 (MHS). The AI analysis found a substantial deficit in granular glycogen content of skeletal muscle in the MHS individuals, more marked for those with high clinical index. Given the limited number of patients in this sample, the conclusion should be taken as preliminary.

## Materials and methods

The goal was to automatically and objectively measure the concentration (number of glycogen granules per unit volume) in different regions of the myofiber. This task requires quantifying granules and evaluating the area and volume of the cell region where they are found. Therefore, artificial neural networks (herein "models") of the convolutional, or CNN, type were implemented for the two tasks.

We attempted first a "categorical classification" (CC) approach, whereby the models assign every image to one of several classes. Because of their substantial limitations, we also developed "semantic segmentation" (SS) models, which, instead of images, classify every individual pixel in them. Both approaches required separate models for implementing the two tasks at hand, resulting in two Locations and two Granules models. The categorical models will be referred to as classifiers and the others as segmenters. All the limitations of the classifiers were overcome by the segmenters. The classifiers, however, provided some results that we found useful to validate the output of the segmenters. Because of the classifiers' limitations, the description

Table 1. **Demographic and clinical data of patients included in this study**

| ID | Age/sex | History/reason for CHCT | CHCT Caff (g) | CHCT Halo (g) | MH status | Clinical signs/symptoms | Histopathology | Clinical index | Fasting glucose (mmol/liter) | Calcium index | RYR1/CACNA1S genetics |
|---|---|---|---|---|---|---|---|---|---|---|---|
| 139 | 43/F | FH | 0 | 0 | MHN | No symptoms | Scattered atrophic polygonal fibers | **1.67** | 4.0 | Not done | Not done |
| 143 | 38/F | Family w' myopathy | 0 | 0.8 | MHS | Heat stroke and cramps | Negative | **6.67** | 5.7 | 7.5 | *RYR1* p.R682Q |
| 144 | 48/M | FH | 0 | 0.5 | MHN | No symptoms | Negative | **0** | 6.1 | Not done | *CACNA1S* p.G1210R |
| 145 | 24/M | Self | 0 | 1.2 | MHS | Cramps and rhabdomyo | Negative | **6.67** | 6.9 | 7.5 | *CACNA1S* p.R1229H |
| 146 | 26/F | FH | 0.5 | 1.8 | MHS | Cramps | Negative | **1.67** | 5.5 | Not done | *RYR1* p.R3502W |
| 173 | 38/M | FH | 0.1 | 3.4 | MHS | No symptoms | Negative | **1.67** | 4.6 | 3.33 | *RYR1* p.Y3540F *CACNA1S* p.T1354S |
| 277 | 51/M | FH | 0 | 0.2 | MHN | No symptoms | Negative | **0** | 5.4 | 6.67 | Negative |

of their structure and outputs is consigned to a supplement (Data S1). Their annotated codes can be found in the GitHub repository (Rios, 2023a, 2023b).

## Semantic Segmentation models

Two segmenter models, Locations and Granules, were built, in both cases starting from the "UNet" structure (Ronneberger et al., 2015, *Preprint*). Given the work required for labeling images for SS, UNet was selected largely for its reported ability to train with few labeled images. The code (written in the Python-Tensorflow-Keras environment) and layer structure of the models, together with pre- and postprocessing coding (done in the IDL environment) are listed in Data S2. The codes are also deposited in GitHub (Rios, 2024a, 2024b).

The UNet structure consists of symmetrical encoding and decoding stages where the image is progressively downsized ("downsampled") by factors of 2 while the information is retained in reciprocally increasing numbers of channels produced by different convolutional filters. For reasons that we did not establish, models with the original UNet structure (Ronneberger et al., 2015, *Preprint*) could not be trained to process our images with high accuracy. By trial and error, we reached the structures used here (detailed in Data S2 ), which achieved the desired accuracy.

## The Locations segmenter

The output of the Locations segmenter is a classification of pixels in 6 classes: unclassified-unknown (label 0), A band (1), I band (2), mitochondria and Z disks (3), SR (4), and near-SR (5, a region that includes all cytosol in inter-myofibrillar spaces). An earlier version, which included separate labels for Z disks and mitochondria, resulted in confusion (both regions are very dark in most images). The consolidation of Z disks and mitochondria had no incidence in the final assignment of granules to locations, as few granules were found in either region (in agreement with the literature, e.g., Marchand et al., 2007).

The segmenters worked well on a dataset that included every available image, regardless of magnification, microscope (three

were used), or even preservation of structure and general quality of the image. This robustness allowed the processing of 462 images. "Locations" was trained on 91 of them, which were chosen at random, including images from every one of the seven subjects in the sample. The training set was enriched by data augmentation (rotations, mirroring, and changes in contrast). The training process alternated optimizers between Adam (Kingma and Ba, 2017, *Preprint*) and Nadam (which features Nesterov accelerated gradient descent or "Nesterov momentum" [Ruder, 2017, *Preprint*]). Validation was carried out on 15 images, and testing on an additional 15. The optimized model reached accuracies of 0.97 in training, 0.82 in validation, and 0.81 in testing.

The performance of the trained Locations model is illustrated in Fig. 2. The top row, A, shows the fitting of images in the training set and the lower row illustrates the performance on the test set. While the fit of the training image is nearly perfect, consistent with the reported 97% accuracy, the predictions on the test images show errors. Over multiple images, the errors largely resided in the transition area between sarcomeric bands A and I, which trainer and model left unclassified (black in the label mask).

## The Granules segmenter

The Granules segmenter was built with a nearly identical structure as Locations (code shared in Data S2 and Rios, 2024b); the necessary difference was in the last classification layer, which had just 2 units (for classifying pixels as "in granule," label 1, or "not," 0). The Granules segmenter was applied to the common dataset of images, using the same sub-set of images for training; data augmentation was used as described for the Locations segmenter. The training process also alternated Adam and Nadam optimizers. Accuracy reached 0.94 in the training set, 0.91 in the validation set, and 0.92 in the test set. The performance is illustrated in Fig. 3. There were predictions of granules where none were present, as well as omissions of actual granules, but both errors were rare. Most of the inaccuracy consisted of a failure to reproduce the abrupt transition at the

| Original image | "Ground Truth" mask | "Predicted" image |

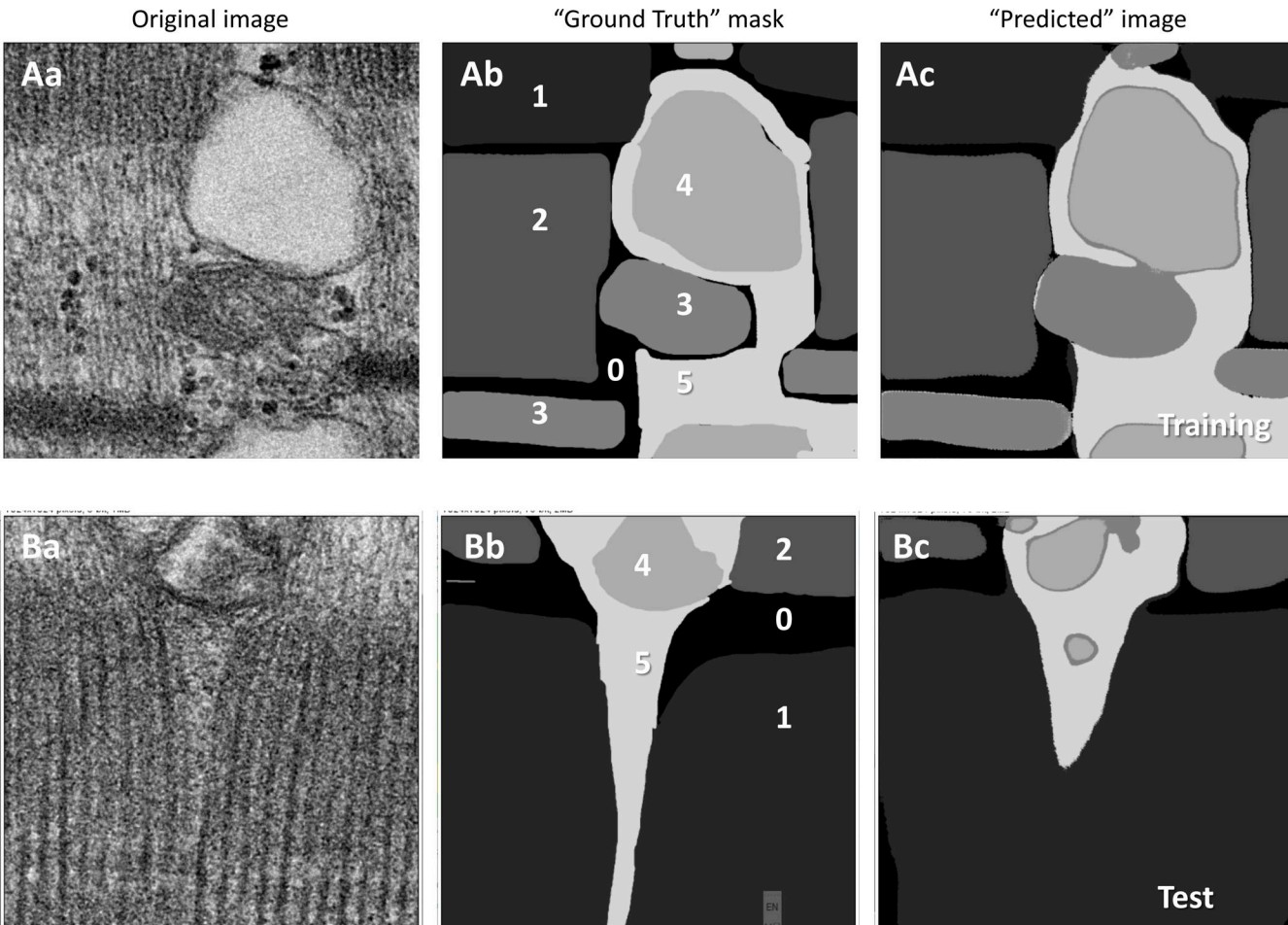

Figure 2.    **Performance of the Locations segmenter. (Aa)** Original EM image. **(Ab)** Label mask provided by the trainer; an array of pixels with label 1 (A band), 2 (I band), 3 (mitochondria or Z disk), 4 (SR), 5 (inter-myofibrillar space excluding mitochondria and SR, "Near SR" in tabulations), and 0 (unclassifiable). **(Ac)** Predictions by the model after training was completed. **(B, a–c)** Same as A, for a test image. Data traceback: A, 0245.png; 035aL-b99-2D16_15k_cell3_s_15. Patient ID 139 HN. B: 037a_b99_7D_16_15k_s_6. Patient 146, MHS.

edge of granules present in the training labels (Ground Truth). To further explore this error, the image regions within the boxes in Fig. 3 are amplified in Fig. 4, together with a plot along the $x$ coordinate at an arbitrary $y$ value. As shown, the error in the predicted image consists of oscillations between 0 and 1 at the label borders and areas in-between close granules. The consequences of these errors for the evaluation of granular content are minor, as argued later.

The operation of the two models working together is illustrated in Fig. 5, which shows an original test image (A) and the segmentation by Locations (B), and Granules (C). The superposition of both outputs (D) allows the derivation of the main quantitative output of the AI analysis. Namely, if $n_R$ = number of pixels in granules in region $R$ and $N_R$ = number of pixels in region $R$, the pixel fraction $pf_R$ (fractional area of granules in region $R$) is defined as

$$pf_R \equiv n_R/N_R \qquad (1)$$

(the subindex $R$ is omitted when referring to cell-wide values).

## Postprocessing

The ultimate goal is to obtain the fraction $vf$ of voxels within granules ($\equiv$ voxels in granules/all voxels), a magnitude thought to be directly proportional to the concentration of glycogen. This measure, of a feature in the 3-D tissue sample, must be derived from $pf$ (Eq. 1), a feature in the 2-D space of EM images. $pf$ and the various $pf_R$ are measured on EM images, which for the purpose of this study are treated as 2-D projections on the $x$-$y$ plane of the granules in the imaged slice. The relationship between $vf$ and $pf$, which in very thin sections should be linear, is not trivial. Nonlinearities arise chiefly from two sources: first, more than one granule fits in the 60-nm depth of a tissue slice. SS is binary; a pixel gets the value 1 whether or not granules overlap at the location. Overlap can be seen, for example, in Fig. 4, A–C. An additional complication arises from the likely presence of incomplete granules, severed at the top and bottom cuts that produce the slice. Thus, a small circle may represent the full projection of a small granule or a "cap," cut from a larger one.

| Original image | "Ground Truth" mask | "Predicted" image |
|:---:|:---:|:---:|

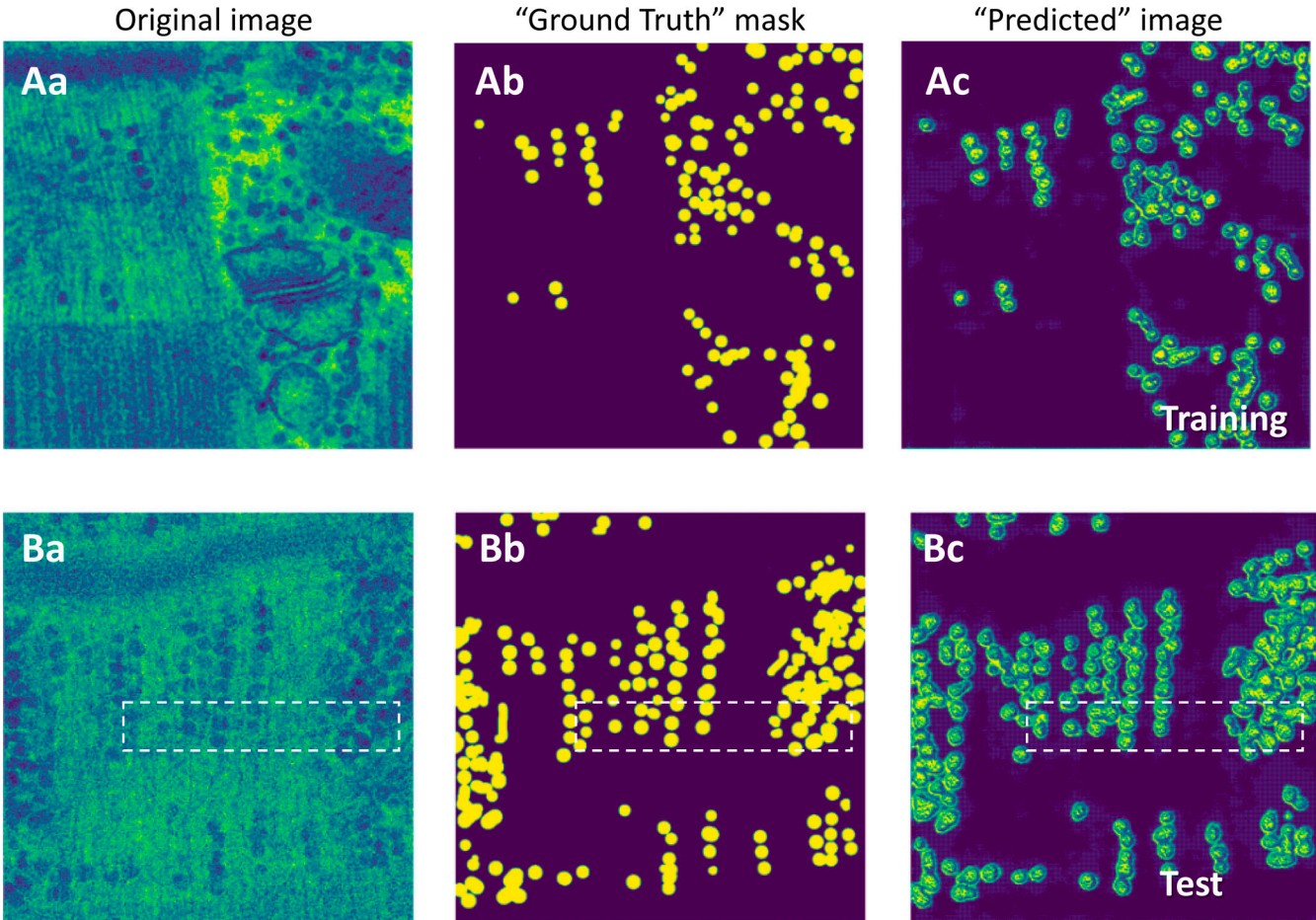

Figure 3.   **Performance of the Granules segmenter. (Aa)** Original EM image. **(Ab)** Label mask provided by the trainer. They are arrays of pixels with value 1 (in granule) or 0 (not in granule). **(Ac)** Predictions by the model, after training was completed. **(B, a–c)** Same as A for a test image. The color palette (provided by a Keras utility) was chosen for better visualization. Boxes mark regions amplified in Fig. 4 in the original grayscale palette. Data traceback: A, 030_grans08_s_5. Patient 144, MHN. B, MS-img-81_s_0. Patient 144, MHN.

## A numerical model for density in granules

While analytical approximations of the relationship between $vf$ and $pf$ were attempted, the most robust solution was found with a numerical simulation of the entire process. Namely, slicing and imaging a virtual volume containing granules of random diameter and location at different concentrations. The output of the simulation (Eq. 2, below) is the formula to derive $vf$, the variable that can be related to glycogen concentration, from the measured $pf$.

The modeling started with the generation of simulated granules in a volume. Because outcomes depend on assumed granule diameters, a realistic simulation requires first measuring the distribution of actual diameters. For this purpose, a computational routine designed by us for identifying and measuring calcium sparks (Cheng et al., 1999; Brum et al., 2000) was adapted to measure granules. As illustrated in Fig. 6, it was applied to the binary "masks" used as labels for training of the Granules segmenter. The figure shows an image (Fig. 6 A) representative of the ones labeled for training, the labels mask (Fig. 6 B), and the contour of granules in the mask (Fig. 6 C). On the contours, the routine measured the lengths (defined as the longest distance between two points in the contour). The utility found 5,168

granules in 91 images. The histogram of their measured lengths is shown in the inset of Fig. 6 D. The peaks at 60 and 80 nm are spurious and found on contours of superimposed granule projections (some are marked by arrows in Fig. 6 C). The lengths shorter than 50 nm were similar to widths in the $x$ and $y$ directions, measured on the same contours, and were therefore taken as measures of diameter. Their histogram is in Fig. 6 D (red), together with those of virtual granules in two separate simulations described below.

In the simulations, a set of 3,000 granules was generated and placed at locations ($x$, $y$, $z$) at uniform random distributions within a basket—a volume of 400 × 400 × 200 discrete locations (referred to as "voxels" or "pixels," respectively in 3-D or 2-D contexts). EM images are all cast at 1,024 × 1,024 0.8 nm pixels. To reduce computational load, the simulation used a 400 × 400 2-D pixel set at 2 nm intervals, as a practical representation of the original image that reduced resolution to 0.4 of the original. Defined in this way, the basket is a version, at 0.4 resolution scale, of a 1,000 × 1,000 × 500 nm physical half-cube (the difference between 1,024 and 1,000 is neglected). In an attempt to reproduce, as closely as possible, the actual distribution of granule sizes, two complete simulations were done:

Figure 4. **Detailed predictions of Granules segmenter. (A–C)** Magnified areas within boxes of Fig. 3 B (original EM, labeled mask, and predictions by the model). **(D)** Pixel intensity along line a-a in B, with values of 1 in granules and 0 elsewhere. **(E)** Intensity along the corresponding line b-b in the predictions mask, showing oscillations at or near the borders of the labeled regions. These oscillations reduce the accuracy of the predictions but (as argued in the text) tend to cancel in the calculation of pixel density. Data traceback: A, 030_grans08_s_5. Patient 144, MHN. B, MS-img-81_s_0. Patient 144, MHN.

one (named Small) with a Gaussian distribution of virtual granule diameters centered at 25 nm and the other (Large) centered at 27.5 nm, both with standard deviations of 5 nm. The histograms of their actual diameters produced with 3,000 instances of a random number generator are the curves in black in Fig. 6 D. Both were used, as alternative representations that bracket the actual distribution (in red).

The simulations placed sets of granules, in increasing numbers, starting from 400 up to 3,000, in eight separate 400 × 400 × 200-voxel baskets (each representing 0.5 µ³ of tissue at this scaled resolution). The concentration $G$ of granules in the different baskets therefore varies between 800 and 6,000 granules per cubic micron, which covers the full range of observed concentrations in different regions (Tables 2 and 3). The set of highest $G$, in its basket, is rendered in Fig. 7 A. From these baskets, x-y slices were cut. To match the actual depth of 60 nm in the scaled-down resolution of the simulation, the slices were 24 voxels in depth. To avoid border effects, slicing was started 12

voxels from the bottom plane. Thus, a total of seven slices were obtained from each basket, as numbered on the ruler in Fig. 7 A. Slice # 0 is represented in Fig. 7 B. As shown, it includes many incomplete granules, severed at the top and bottom surfaces. The binary x-y projection is displayed in Fig. 7 C; at this high concentration, it exhibits multiple silhouettes of superposed granules. Therefore, the model reproduced overlap and granule slicing, the two features that hinder the quantitative extrapolation from EM images.

From the projections represented in Fig. 7 C, a pixel fraction $pf$ was computed (Eq. 1) and the result averaged over the seven slices. Obtained thus, $pf$ is the simulated version of the main quantitative outcome of the AI analysis. One such value is obtained for each of the eight baskets of increasing $G$; the relationship of $pf$ and $G$ is represented with black symbols in Fig. 8; circles for the simulation with larger granules and triangles for the smaller ones. While the dependencies are close to linear, there is a plus deviation that increases with $pf$, a consequence of

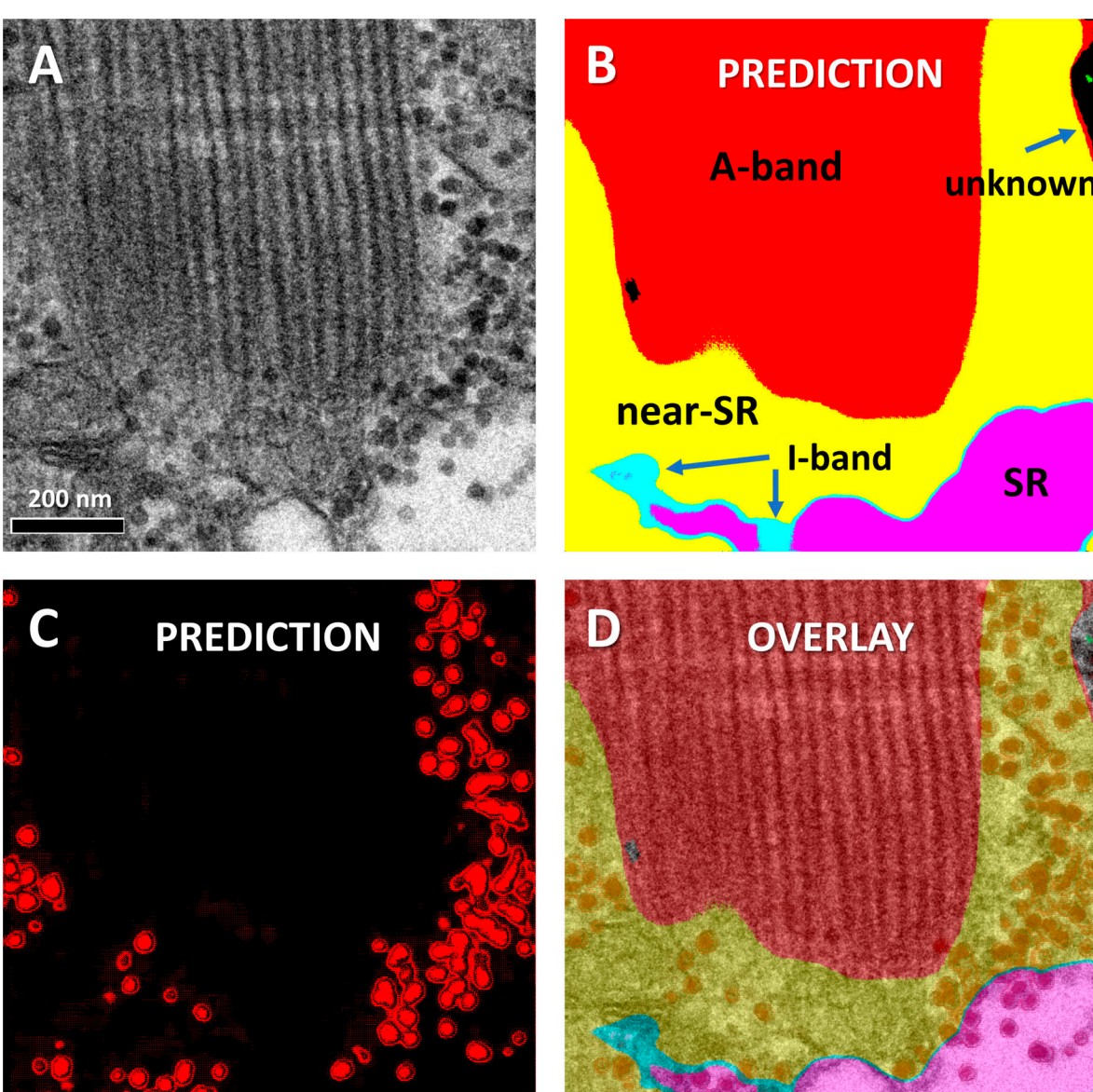

Figure 5. **Location and Granules segmenters working together. (A)** A test EM image of a section slightly oblique to the fiber axis. **(B)** Predictions by the Locations model. **(C)** Predictions by the Granules model. **(D)** Overlay of all panels. Pixel fraction *pf* is calculated as the ratio of pixels in granules over total pixels, in every region and in the total area where the Locations prediction was not 0. Data traceback: A, 030_grans08_s_0. Patient 144, MHN.

the increasing superposition of granules in the projection. The simulation also yields the desired concentration of voxels in granules, *vf*, as the ratio of non-zero voxels over total voxels in the basket. The relationship between this variable and *pf* is plotted with red symbols in Fig. 8, again for the two simulations.

The relationship between *pf* and *vf* is the main product of the simulation. As can be seen in the graph, both *vf* and G depend almost linearly on *pf*. While *G(pf)*, the concentration of granules at a given measured pixel fraction, is predictably lower in the simulation with greater granules (plots in black), that of voxel fraction, *vf(pf)*, barely changes between the two simulations (plots in red). This is a reassuring outcome; it indicates that the extrapolation of measured pixel fraction to fractional volume will be robust in the presence of varying distributions of granule sizes.

Therefore, an analytical formula can be adjusted with confidence to the dependence derived from the simulation. The dependences are nearly linear; a second-order polynomial of the form $vf = a \times (pf)^2 + b \times pf + c$ fit to the *vf* versus *pf* data in the figure gives the values $a = 0.0547$, $b = 0.2448$ and $c = 0.0$, for the simulation with smaller granules (red triangles in the figure), and $a = 0.0534$, $b = 0.2629$, and $c = 0.0001$ for the larger granules (red circles). Because both fits are similar and both distributions of diameters in the simulation approximate the measured distribution similarly, the two sets of values were fitted together, with the result $a = 0.1511$, $b = 0.2344$, and $c = 0.0005$ (fit in Fig. 8 inset). These are the preferred parameter values in the formula that derive fractional volume occupied by granules from measured *pf*:

$$vf = 0.1511(pf)^2 + 0.2344pf + 0.0005. \qquad (2)$$

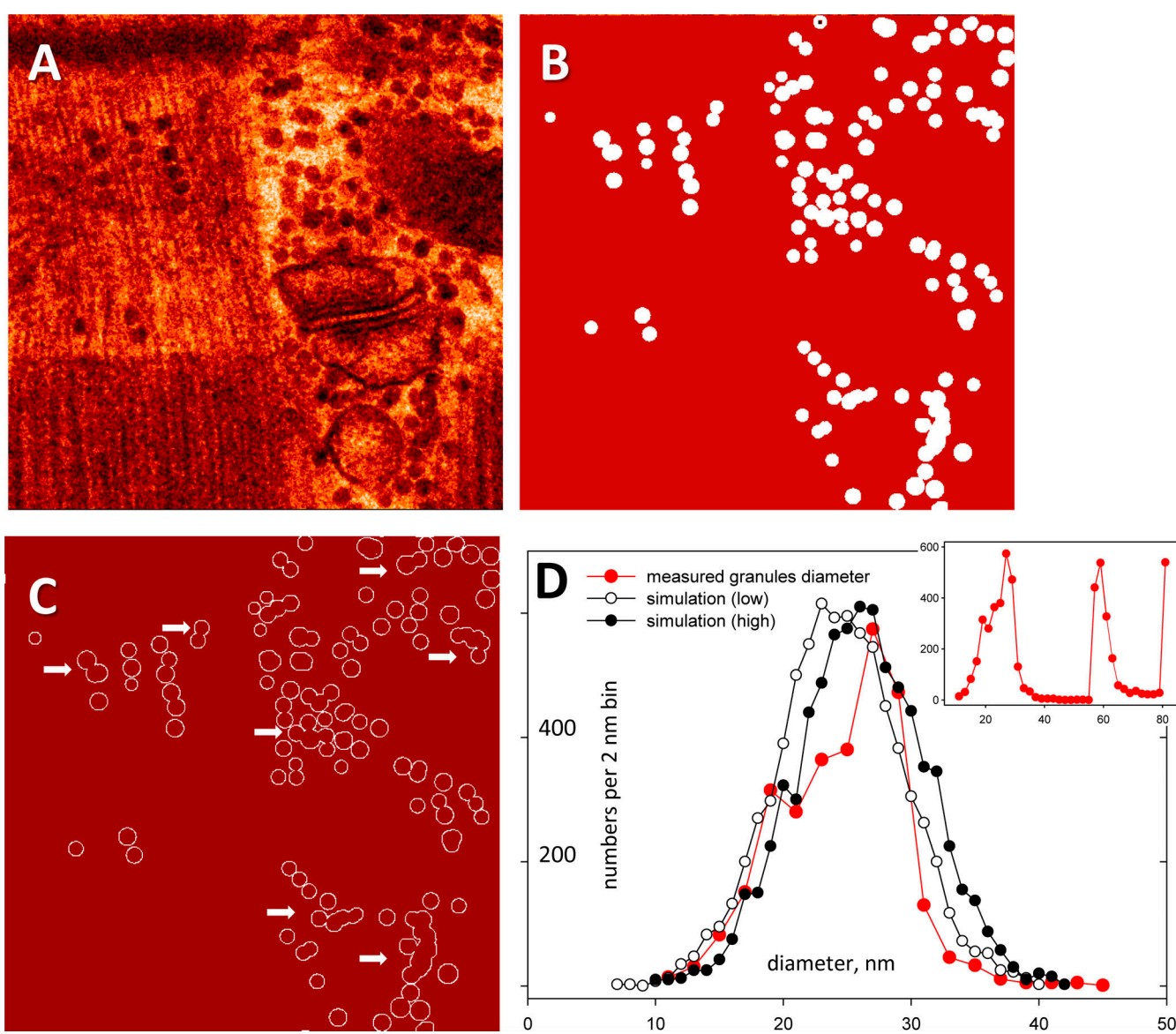

Figure 6.   **The distribution of granule diameters. (A)** An EM image used for testing. **(B)** Binary label mask of granules, provided by a trainer. **(C)** Contour of granules by a computational utility adapted from Brum et al. (2000) and Cheng et al. (1999), with code shared in Data S2. **(D)** Inset: Histogram of lengths, measured by the utility on all contours in 91 labeled images. **(D)** Plot in red: Same histogram with abscissa limited to lengths below 50 nm, representing the distribution of well-measured diameters. The peaks at larger values correspond to lengths on contours of overlapping granules (examples marked by arrows in C). Plots in black: Histograms of 3,000 random diameters with a normal distribution of mean 25 nm (open symbols) or 27.5 nm (filled) and SD of 5 nm. Data traceback: A, A, 030_grans08_s_5. Patient 144, MHN. Graphs in granule numbers and densities in slices. JNB. Section 5.

The simulations also provide two different estimates of the concentration of granules.

$$G = 17{,}571(pf)^2 + 26{,}253pf + 8 \qquad (3)$$

and

$$G = 13{,}940(pf)^2 + 20{,}667pf + 8, \qquad (4)$$

respectively, for the Small and Large simulations.

**From granules to glycogen concentration**

The calculation requires knowledge of the relative molecular mass per glycosyl residue. A range of estimates is found in the literature, ranging from 162 Da (the difference between the $M_r$ of glucose, ~180 Da, and that of $H_2O$, removed stoichiometrically in every condensation reaction) and 181 Da (derived from structure models [Goldsmith et al., 1982] where granules of 42 nm diameter contain ~55,000 glycosyl residues [Roach et al., 2012; Liu et al., 2021] and are thought to have a mass of $10^7$ Da [Prats et al., 2018]). The uncertainty could reflect the presence, variable and not fully understood, of glycogen enzymes and the core protein glycogenin at a stoichiometry of one per granule (Alonso et al., 1995). None of the conclusions reached in the present study would vary if choosing a value within this range. In the calculations below, we use 167 Da, a number reached in two ways: it is approximately one-fourth of the $M_r$ of the branched tetramer of glucose (Human Metabolome Database

Table 2.  **Distribution of granules per region and subject**

| Condition | ID # | Variable | A band | I band | Z disk & Mitoch | SR | Near SR | All regions |
|---|---|---|---|---|---|---|---|---|
| **MHN** | 139 | $pf_R$ | 0.008 | 0.021 | 0.024 | 0.012 | 0.073 | 0.023 |
| | | volume, $\mu^3$ | 1.29 | 0.78 | 0.18 | 0.24 | 0.46 | 2.94 |
| | | $G_R$ small | 210 | 546 | 622 | 315 | 1954 | 608 |
| | | $G_R$ large | 173 | 448 | 510 | 259 | 1599 | 499 |
| | | $vf_R$ | 0.002 | 0.005 | 0.006 | 0.003 | 0.019 | 0.006 |
| | 144 | $pf_R$ | 0.025 | 0.152 | 0.063 | 0.113 | 0.24 | 0.1 |
| | | volume, $\mu^3$ | 2.44 | 1.61 | 0.26 | 0.09 | 0.77 | 5.16 |
| | | $G_R$ small | 650 | 4264 | 1660 | 3087 | 7081 | 2709 |
| | | $G_R$ large | 533 | 3481 | 1358 | 2522 | 5771 | 2214 |
| | | $vf_R$ | 0.006 | 0.04 | 0.016 | 0.029 | 0.065 | 0.025 |
| | 277 | $pf_R$ | 0.038 | 0.036 | 0.106 | 0.075 | 0.114 | 0.073 |
| | | volume, $\mu^3$ | 0.94 | 0.34 | 0.32 | 0.34 | 0.85 | 2.79 |
| | | $G_R$ small | 1004 | 950 | 2883 | 2001 | 3105 | 1934 |
| | | $G_R$ large | 822 | 778 | 2356 | 1636 | 2537 | 1582 |
| | | $vf_R$ | 0.01 | 0.009 | 0.027 | 0.019 | 0.029 | 0.018 |
| **MHS** | 146 | $pf_R$ | 0.004 | 0.01 | 0.037 | 0.014 | 0.137 | 0.026 |
| | | volume, $\mu^3$ | 0.89 | 0.43 | 0.06 | 0.15 | 0.26 | 1.78 |
| | | $G_R$ small | 114 | 262 | 975 | 377 | 3807 | 679 |
| | | $G_R$ large | 95 | 216 | 799 | 310 | 3109 | 557 |
| | | $vf_R$ | 0.001 | 0.003 | 0.009 | 0.004 | 0.036 | 0.007 |
| | 143 | $pf_R$ | 0.005 | 0.018 | 0.038 | 0.023 | 0.072 | 0.021 |
| | | volume, $\mu^3$ | 1.01 | 0.27 | 0.09 | 0.12 | 0.32 | 1.81 |
| | | $G_R$ small | 125 | 470 | 988 | 596 | 1925 | 553 |
| | | $G_R$ large | 104 | 386 | 810 | 489 | 1575 | 454 |
| | | $vf_R$ | 0.002 | 0.005 | 0.01 | 0.006 | 0.018 | 0.006 |
| | 173 | $pf_R$ | 0.007 | 0.037 | 0.031 | 0.048 | 0.122 | 0.035 |
| | | volume, $\mu^3$ | 1.21 | 0.72 | 0.19 | 0.04 | 0.38 | 2.53 |
| | | $G_R$ small | 180 | 976 | 798 | 1260 | 3364 | 909 |
| | | $G_R$ large | 149 | 800 | 654 | 1031 | 2748 | 745 |
| | | $vf_R$ | 0.002 | 0.009 | 0.008 | 0.012 | 0.031 | 0.009 |
| | 145 | $pf_R$ | 0.002 | 0.008 | 0.015 | 0.022 | 0.026 | 0.007 |
| | | volume, $\mu^3$ | 0.52 | 0.44 | 0.09 | 0.04 | 0.09 | 1.18 |
| | | $G_R$ small | 60 | 202 | 385 | 569 | 676 | 196 |
| | | $G_R$ large | 51 | 167 | 317 | 467 | 555 | 162 |
| | | $vf_R$ | 0.001 | 0.002 | 0.004 | 0.006 | 0.007 | 0.002 |

Variables calculated by Eqs. 1, 2, 3, and 4. Averages over conditions are listed in Table 3. Traceback: raw data in Areas per region. JNB. Section 7, number of granules and density per micron. Conversion $pf$ to $vf$ and G.xlsx.

accession no. HMDB0000757; [National Center for Biotechnology Information, 2024], assumed to be the typical subunit in glycogen; it is also equal to the mass per residue of a large linear or branched homopolymer of glucose) increased by 3% to account for the presence of proteins (reviewed by Bezborodkina et al. [2018] and Prats et al. [2018]).

These estimates put $[\text{glycogen}]_G$, the glycogen concentration in grams per liter of large granules at:

$$[\text{glycogen}]_G = \frac{55{,}000 \frac{\text{glycosyls}}{\text{granule}}}{\frac{4}{3\pi} \times \frac{21^3 \text{nm}^3}{\text{granule}}} \times \frac{10^{24} \text{nm}^3}{l} \times \frac{167 \text{gm/mole}}{6.023 \times 10^{23} \text{glycosyl/mole}}$$

$$= 393 \text{gm/liter}. \tag{5}$$

Assuming that $[\text{glycogen}]_G$ is independent of granule size, the concentration of glycogen per liter of cell is

**Table 3.  Distribution of granules grouped by diagnostic condition**

| Condition | Variable | A band | I band | Z disk & Mitoch. | SR | Near SR | All regions |
|---|---|---|---|---|---|---|---|
| **MHN** | $pf_R$ | 0.023 | 0.101 | 0.072 | 0.057 | 0.151 | 0.072 |
| | volume, µ³ | 4.67 | 2.73 | 0.76 | 0.67 | 2.08 | 10.89 |
| | $G_R$ small | 598 | 2733 | 1916 | 1511 | 4237 | 1911 |
| | $G_R$ large | 491 | 2234 | 1567 | 1237 | 3459 | 1563 |
| | $vf_R$ | 0.006 | 0.026 | 0.018 | 0.014 | 0.039 | 0.018 |
| | [Glycogen], g/l | 2.334 | 10.114 | 7.002 | 5.446 | 15.171 | 7.002 |
| **MHS** | $pf_R$ | 0.005 | 0.021 | 0.03 | 0.022 | 0.102 | 0.025 |
| | volume, µ³ | 3.63 | 1.86 | 0.43 | 0.35 | 1.05 | 7.3 |
| | $G_R$ small | 131 | 556 | 774 | 571 | 2772 | 648 |
| | $G_R$ large | 108 | 456 | 635 | 468 | 2266 | 531 |
| | $vf_R$ | 0.002 | 0.006 | 0.008 | 0.006 | 0.026 | 0.006 |
| | [Glycogen], g/l | 0.778 | 2.334 | 3.112 | 2.334 | 10.114 | 2.334 |

$vf$ and $G$ calculated by Eqs. 1, 2, 3, and 4. [Glycogen] by Eq. 7. Traceback: raw data in Areas per region.JNB, Section 12. Exported to: Conversion $pf$ to $vf$ and G.xlsx.

$$[glycogen]_c = [glycogen]_G \, vf = 393 \, vf \text{ gm/liter of cell.} \quad (6)$$

Using 9% as a measure of extracellular volume (Banypersad et al., 2013) yields concentration in muscle as

$$[glycogen]_M = 358 \, vf \text{ gm/liter of tissue,} \quad (7)$$

with $vf$ derived from the measured $pf$ via Eq. 2.

### Fixation, embedding, and imaging

Human samples are remnants from muscle biopsies collected for diagnostic purposes from patients of the MHIU of the University Health Network at the Toronto General Hospital. Patients signed forms that included consent for further use of their tissue in the present research. Protocols were approved by the Institutional Review Boards of the Universities of Toronto, and Rush and Virginia Commonwealth Universities, where imaging took place. Materials were shipped overnight at 4°C in a low [Ca²⁺] relaxing solution described by Figueroa et al. (2019). Fiber bundles were mounted moderately stretched in the relaxing solution on Sylgard-coated dishes. The solution was then replaced by a fixative containing 4% PFA for 20 min.

Small bundles of fibers were rinsed at least three times in 0.1 M cacodylate buffer, stained in 2% osmium tetroxide plus 0.8% potassium ferrocyanide in 0.1 M cacodylate buffer for 1 h at room temperature, then contrasted with saturated uranyl acetate for 1–2 h at room temperature in the dark. After multiple dehydration steps, the tissue was embedded in Embed 812 resin overnight at 60°C. Thin sections of 50–70 nm were cut with ultramicrotome, stained with lead citrate, and viewed on a Tecnai F20 (Thermo Fisher Scientific) electron microscope at Virginia Commonwealth University, a Tecnai F20 at the University of Virginia and a JEM-1010 (JEOL USA INC.) at the University of Pennsylvania. Most tissues were processed in parallel without potassium ferrocyanide in the post-fixation solution.

### Preprocessing

For SS, images acquired originally at variable magnifications and digitized in a range of 0.4–1.4 nm pixel distance were recast as 1,024 × 1,024 pixels at 0.8 nm. 91 images were labeled (annotated) for both Granules and Location segmenters. Labeling was carried out by four different trainers after developing common criteria and approved by a final trainer, who could also request revisions. Labeling was done using Labkit (Arzt et al., 2022) in the FIJI version (Schindelin et al., 2012) of the ImageJ environment (Schneider et al., 2012). Label masks were output as byte grayscale arrays in PNG format. The training data, images, and label masks were entered using the Dataset procedure of the Tensorflow Keras environment (https://github.com/tensorflow/tensorflow/releases/tag/v2.11.0).

### Postprocessing

The model predictions, output as PNG arrays of the same dimensions as the images and label masks, were combined computationally to yield, for every image, a list (vector) of pixel count pairs—pixels within granules and total pixel count—for every cell region (regions listed at top of Table 2). From these, a volume was calculated (in µ³) and, using Eq. 1, a $pf_R$ value for every region and a $pf$ for the whole segmented area of the image.

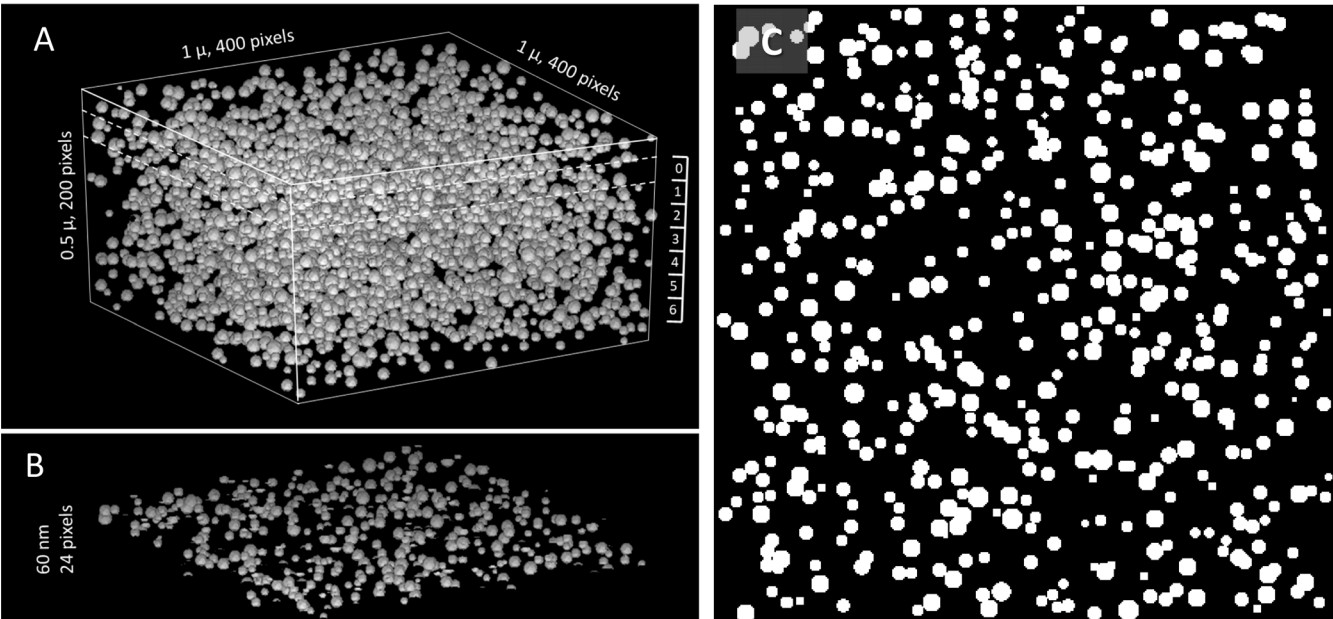

Figure 7. **A simulation of granules in a volume. (A)** 3,000 granules with the distribution of diameters represented by their histogram in Fig. 6 D (black filled circles) at random locations within a half-cube of the dimensions indicated, simulating a concentration of 6,000 granules per μ³. **(B)** A slice of 60 nm equivalent thickness (24 voxels-deep), cut from the volume in A at *x-y* planes indicated by dashed lines and identified as "0" on the ruler at right in A. **(C)** Two-valued (0, 1) projection on the *x-y* plane of the slice in B. The projection simulates the output of the Granules segmenter working on an EM image. Its *pf* is calculated as the fraction of pixels in granules. The *pf* corresponding to this concentration of granules is obtained as the average of *pf* values in seven similar slices extracted from the volume as indicated by the ruler in A. Similar calculations were carried out for eight concentrations of granules and two distributions of diameters, with results presented in Fig. 8. Simulation code listed in Data S2.

The variables *vf* and *G* were derived from *pf* (Eqs. 2, 3, and 4), and the concentration of glycogen was calculated from *vf* (Eq. 7). These computations were implemented as programs in the IDL environment (Harris Geospatiale). Variables calculated in individual images were averaged for individual patients and then grouped by condition (MHN versus MHS) or according to clinical index. Simple statistical comparisons (*t* tests and Kruskal–Wallis Rank Sum tests) were done in Sigmaplot or Excel. Hierarchical statistical comparisons were carried out following the protocol described in Sikkel et al. (2017) and Knowles (2013), implemented in the RStudio interface (RStudio, 2011) of the R language environment (GNU General Public License, 2024).

## Results

492 images of 1,024 × 1,024 0.8-nm pixels were analyzed (including the images used for training, validation, testing, and prediction) from three MHN and four MHS subjects. The volumes of tissue processed were nearly 11 μ³ from the MHN subjects and 7.3 μ³ from the MHS patients. The predictions of the Locations and Granules segmenters working on the same image are illustrated in Fig. 5. The quantitative outputs $N_R$ and $n_R$—the numbers of pixels of six classes assigned by the Locations segmenter and two classes by Granules—were ratioed to yield *pf* (Eq. 1), from which the volume density *vf* and concentration of granules *G* were derived (Eqs. 2, 3, and 4). These are graphed per image and subject in Fig. 9 and replotted as a function of clinical index in Fig. 10.

The main results for the MHN subjects are as follows: the greatest fractional volumes were occupied by A and I bands, and intermyofibrillar cytosol (∼40, 20, and 15%, respectively). By Eq. 2, the fraction of pixels in granules (*pf*) translated to a volume fraction *vf* of 0.018, close to 2%. Eqs. 3 and 4 defined a range between 1,563 and 1,911 for the number of granules per cubic micron. The highest concentration, 3,459 to 4,237 granules μ⁻³, corresponding to a volume fraction of ∼4%, was found in the intermyofibrillar cytosol ("Near SR" column in tables).

The results for MHS patients listed in Table 3 include a *vf* of 0.006, which is threefold lower than that in the MHN. The large difference prevails in every region of the cells and correlates with the severity of the syndrome. The hierarchical statistical analysis of the significance of differences for three comparisons is documented in Table 4. Granule fractions of MHN and MHS are significantly different (P ∼0.045). Those of the MHS patients with a manifest disease (clinical index value of 6.67, in red in Fig. 9) are significantly different from the patients with the normal phenotype (clinical index = 0, P = 0.035), while the difference with the patients with intermediate phenotype barely misses the 0.05 threshold. Since we know, from biochemical analyses, that MHS patients store less glycogen (e.g., Tammineni et al., 2020), a Bayesian argument can be made for the one-tail evaluation of the first-class error, which would double the significance of the differences. The negative correlation of granule fraction and clinical index (P << 0.0001) is clearly shown with Fig. 10.

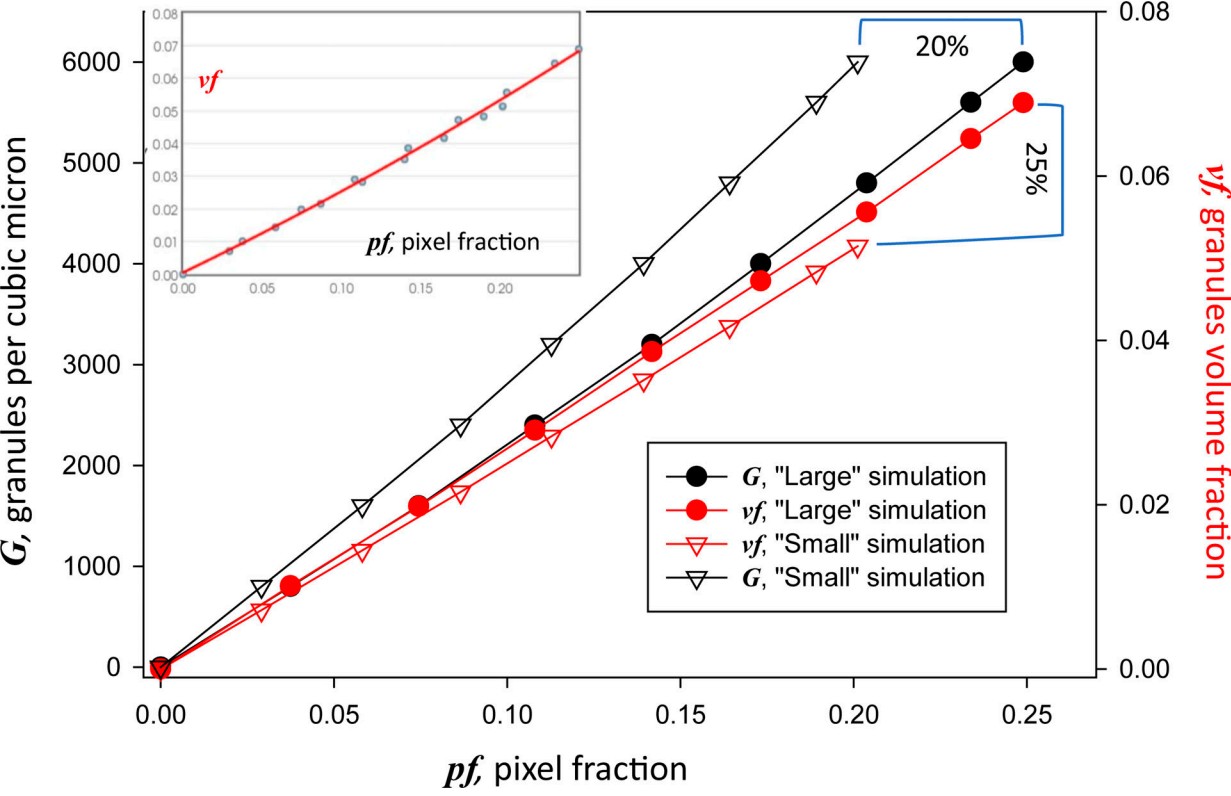

**Figure 8. Numerical relationship between pixel fraction in images and variables in the volume.** In abscissa, pixel fraction, measured in eight baskets for each of two simulations: Small, with granule diameters centered at 25 nm, triangles, and Large, centered at 27.5 nm, circles. Red symbols plot values of *vf* derived from the simulations as described in Fig. 7. The horizontal and vertical brackets mark the maximal difference in *pf* and that in *vf*, respectively, between the two simulations. Black symbols plot the concentration *G* of granules in the respective baskets. As expected, *G(pf)* functions differ, as higher concentrations are required to reach a given pixel fraction with the smaller granules. In contrast, *vf* (in red) is nearly insensitive to the size of granules. All dependencies are nearly linear and were well described by second degree polynomials (Eqs. 2, 3, and 4). The *vf* (*pf*) data were collectively fit by a single polynomial (Eq. 2 and inset). Data traceback: Granule numbers and densities in slices. JNB. Section 1.

### Segmenters versus categorical classifiers

In view of their limitations, the categorical models were applied only to images obtained in one microscope at one magnification. They were from five patients, four with a positive MHS diagnosis and one with MHN. The output of the Locations and Granules classifiers, separately for regions and whole images, is presented in Data S1; and Tables 1 and 2.

Samples from the five individuals processed with the categorical classifiers were also analyzed by the segmenters. While neither the images nor the location classes used were exactly the same for the two approaches, we compared the results to rule out major calculation errors. The test seemed especially necessary for the estimates derived from the segmenters, as the numerical simulation that underlies those estimates added a possible source of errors.

The comparison of granule concentrations *G* derived with the two approaches is illustrated with the cross-plot in Fig. 11. In ordinates are the values obtained by the classifiers; in abscissas, those by the segmenters, which are two for each subject, corresponding to the simulations named Small, represented by circles, and Large, by squares. A *t* test of the paired differences categorical-segmenter yielded an average of +73.2 or 14%, a P = 0.274 of difference due to chance and a 95% confidence interval

−69–215. The power of the test, however, was small (0.08). In sum, the comparison provided no indication of major quantitative errors in the segmentation approach, but the test cannot exclude systematic biases in either method.

### Discussion

The technical advance communicated here is the construction of artificial neural networks to accomplish two complementary tasks: quantification of glycogen granules and their assignment to different regions within EM images of "ultrathin" sections of muscle cells. (In postprocessing that did not entail AI, the separate classifications were then combined to obtain estimates of concentrations of granules and volumes occupied by them in the locations of interest.)

Initially, we used a categorical classification approach, which starts from the subdivision of the large EM images into small subimages. The subdivision not only simplified counting (by limiting the maximum number of granules) but also reduced the problem of locating granules to a simple additional classification of images pertaining to different regions. While conceptually simple, the approach found multiple limitations; it proved unsuitable for the classification of images acquired at different

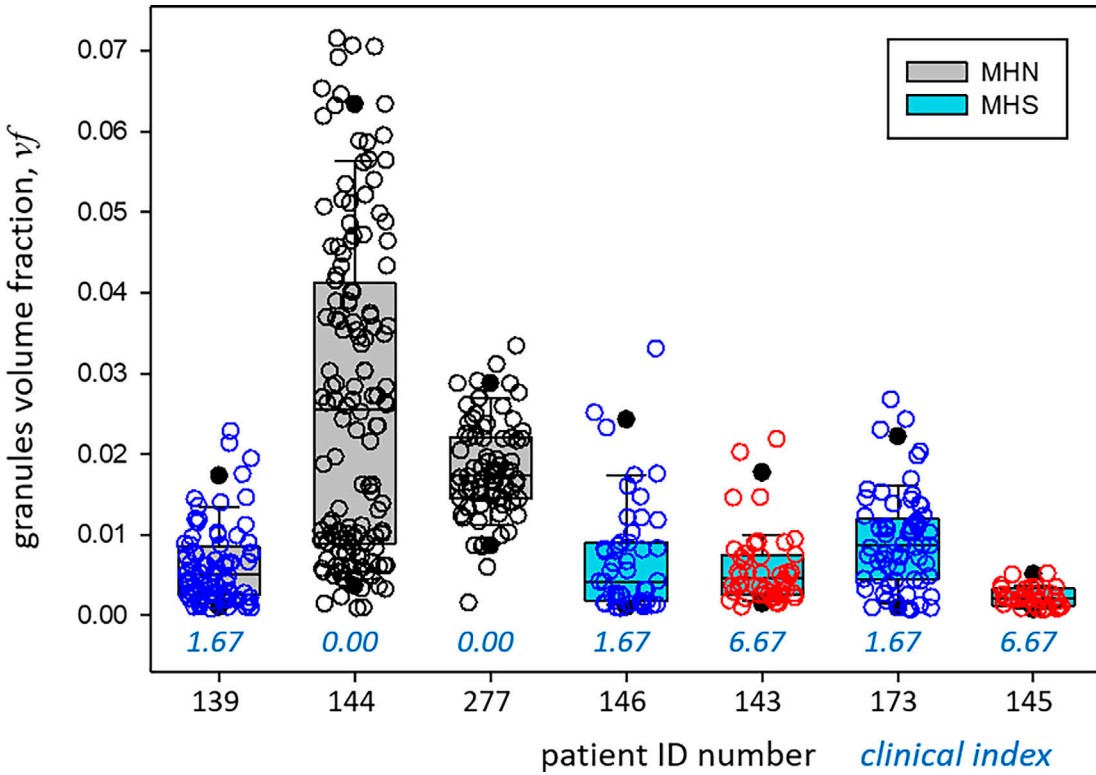

**Figure 9.** **The fractional volume occupied for granules in seven patients.** Values of *vf* are plotted individually for every image (circles) and collectively as box plots (mean, 25 and 75% limiting the box, 5 and 95% at circles). The circle colors code the patient's clinical index (black: 0.00; blue: 1.67; red: 6.67). The statistical comparison of MHN and MHS patients is documented in Table 4. Data traceback: Granule density box by seven patients.JNB. Section 1.

magnifications and with different hardware. Most importantly, it proved incapable of actual "counting," i.e., distinguishing numbers of granules in increments of one; the approach became acceptably accurate only when counts were binned, so the precision of the quantification was coarser than one.

The limitations of the categorical classifiers were overcome by the semantic segmentation approach; the segmenters could be applied to sets of images acquired at any resolution between 0.4 and 1.4 nm/pixel, with different microscopes, and tolerated wide variations in image quality and structure conservation. In testing, the Locations and Granules segmenters reached an accuracy (fraction of correctly classified pixels) of 81 and 92%, respectively. If the errors in Locations and Granules were independent, the concentration measure resulting from the combination of the two outputs would be accurate only in 75% of cases (0.81 × 0.92). However, as argued with Fig. 4, most of the inaccuracies of the granules segmenter cancel in the final calculation. Similarly, most errors of location occurred in areas that were unclassifiable and were given the label "0," which removed them from the calculation of fraction, thus not adding to its errors. In conclusion, we believe that the present calculations of pixel fraction in granules are accurate to better (and arguably much better) than 75%.

As detailed in Data S1, the categorical classifiers reached similar levels of accuracy, but only in a subset of images. A wide exploration of different network structures, described in the supplemental data, suggests that the CC approach that we developed is intrinsically worse than SS for the intended task. We propose, without proof, that the approach, based on the subdivision of the EM images into small sub-images, loses the spatial context—the surroundings—that the full images provide and is probably key to correct assignments in either approach.

The semantic segmenters produce as output *pf*, the fraction of all pixels occupied by granules, in a region or in the whole segmented area of the image. This is a measure of the EM image—a two-dimensional object. The derivation from it of the volume fraction *vf* occupied by granules, a measure of a three-dimensional object, was reached via a detailed numerical simulation of the granule-containing tissue, illustrated with Figs. 6, 7, and 8. Fig. 8 represents an outcome of the simulation: the numerical relationship between *vf* and *pf* (plots in red) separately for the two simulations with different distributions of granule diameters. Additionally, the simulations produced the concentration *G* of granules (in black). These plots reveal an unexpectedly virtuous outcome: while the functions *G*(*pf*) depend strongly on granule size (fewer large granules produce the same pixel content as greater numbers of smaller ones), the function *vf* (*pf*) is nearly insensitive to granule size. This robustness increases confidence in the derivation, as it becomes nearly independent of any difference in the size of virtual and real granules. A final bonus is that a single analytical formula (Eq. 2) can be used to derive *vf* from *pf*.

Eqs. 3 and 4 provide a range of values for *G*, the concentration of granules (numbers per cubic micron), a variable of interest for comparison with the *G* emergent from the categorical analysis. The conclusions from this comparison are that the

**Figure 10. The fractional volume occupied for granules correlated with the clinical index.** Values of *vf* are plotted individually for every image (circles) and collectively as box plots grouped according to clinical index. The correlation coefficient $r^2$ for 487 equal-weighted images is −0.437, with P << 0.0001. Hierarchical statistical comparisons are documented in Table 4. Traceback: Granule density box by seven patients.JNB. Section 1.

observed differences (lower values of *G* in the segmentation approach), while not significant statistically, do not rule out a systematic difference, which would reflect errors in either approach. The magnitude of the differences (14% on average) rules out major errors in the actual measurements and the post-processing calculations.

**Other possible errors**

A critical examination of the procedure leading to *vf* and glycogen concentration reveals one weakness. The *vf* (*pf*) relationship (Eq. 2) emerges from a simulation of granules that copies the measured distribution of granule diameters. The measurement of these diameters, however, is done on EM

Table 4. **Pairwise comparisons of granule density by hierarchical analysis (Sikkel et al., 2017)**

| 1 | 2 | 3 | 4 | 5 | 6 |
|---|---|---|---|---|---|
| **Condition** | **subjects** | ***vf*** | **Clustering (ICC)** | **Corrected Std error** | ***P*** |
| **MHN** | 139, 144, 277 | 0.0174 | 19.70% | 0.005 | 0.0447 |
| **MHS** | 146, 143, 173, 145 | 0.006 | | | |
| | | | | | |
| Clin Index high, (6.67) | 143, 145 | 0.0042 | 4.20% | 0.004 | 0.0035 |
| Clin Index = 0 | 144, 277 | 0.0226 | | | |
| | | | | | |
| Clin Index high, (6.67) | 143, 145 | 0.0042 | 5.00% | 0.001 | 0.0537 |
| Clin Index med, (1.67) | 139, 146, 173 | 0.074 | | | |

Columns: (1) conditions compared, by pairs; (2) subjects included in the compared groups; (3) averages of density over equal weighted images; (4) clustering in data, quantified by the intraclass correlation coefficient ICC; (5) SE of difference corrected for clustering; (6) P value, corrected for clustering. Comparisons by diagnostic status are illustrated in Fig. 9 and comparisons by clinical index are in Fig. 10. Traceback: raw data in areas per region.JNB Section 9, exported to granule_densities_for _hierarchical _[HN_HS, Clin0_Clin667, Clin167_Clin667].xlsx."

Figure 11. **Comparison of concentration of granules, G, calculated by two AI approaches.** Results in images from five human subjects. CC provided one estimate for each subject (in ordinates). SS provided a range between a low and a high value (in abscissas), corresponding to the two distributions of granule sizes used in the simulations leading to Equations and 4. Values for the single MHN individual are plotted with black symbols. Other patient data are provided in Table 1. Data traceback: Granule density categorical versus semantic JNB.

images (strictly, on labels, i.e., silhouettes derived from granules). But, as argued, in EM images, small granules cannot be distinguished from horizontally cut "caps" of bigger granules. Thus, the measured diameters bias the actual distributions toward smaller values. This could be a reason for the skew in the measured curve (Fig. 6 D). The bias implies that our simulations assume granule diameters that are too small. The error is correctible (specifically, it should be reflected in the divergence between the distribution in EM images and that in the *x-y* projections of simulated slices, a difference that could be minimized by iteratively adjusting the simulations until the best match between measurements in EM and virtual slices is reached). This was not done, both because of its computing "overhead" and because the outcome (the *vf* (*pf*) function) proved insensitive to variations in the diameters of the virtual granules.

Inspection of the EM images in Figs. 1 and 3 reveal another kind of mismatch with the simulations, which model the spatial distribution of granules as random. While in EM images granules appear randomly distributed in inter-myofibrillar spaces, those within myofibrils are aligned with the myofilaments. The errors in the analysis due to such discrepancy are probably minor because the granular content within myofibrils is low, and, as shown, proportionality between *vf* and *pf* is an excellent approximation at low fractions.

### Implications for physiology and disease

The concentration of glycogen in granules derived from the segmentation analysis was ~6.5 g/liter of muscle on an average in the three MHN patients studied (Table 3). Excluding from the average the result for patient #139, which could be seen as an outlier because it is typical for MHS patients, the number rises to ~8 g/liter. Compared to a total content of glycogen of 20–30 g/liter of muscle in sedentary subjects at rest (Burke et al., 2017; Marchand et al., 2007), the present estimate leaves a large margin for glycogen content in other forms. The large excess remains after adding contributions to granular content from subsarcolemmal areas, which were not sampled in the images analyzed but could add up to 10% of total glycogen (Marchand

et al., 2007; Ørtenblad and Nielsen, 2015). Larger counts of granules found in previous studies (e.g., 2,700 μm⁻³ [Marchand et al., 2007]) would still leave a substantial difference. As noted above, some of the glycogen-stained EM images show remarkably thick staining of SR membranes, indicating glycogen bound to their cytosolic surface. This location is also consistent with the preferential location of GP, the rate-limiting enzyme of glycogenolysis, shown by Tammineni et al. (2020) to be present in SR membranes.

The present study finds the highest concentration of granules in the intermyofibrillar space. The finding is qualitatively consistent with the summary data in the review by Ørtenblad and Nielsen (2015). However, our results diverge, in the substantial content of granular glycogen found in the intrafibrillar space, especially within the I band (Tables 2 and 3, column 5). Given the limited sample explored here, we can only report the discrepancy.

The study revealed a threefold deficit in granular content in patients with the MHS condition; the difference was statistically significant. The result is qualitatively consistent with the report of lower glycogen content measured by chemical analysis in muscles of 13 MHN and 12 MHS subjects (Tammineni et al., 2020), even though the analyses were not strictly comparable because Tammineni's was done on microsomal fractions instead of whole-muscle samples.

MHS is not regarded as a disease but as a "condition," as often their subjects do not have overt impairment of function or symptoms of disease. The clinical manifestations of MHS, however, can be significant, so that Figueroa et al. (2019) found it useful to subsume their joint import in a quantitative clinical index. This index was consistently elevated in the MHS and correlated with a calcium index, which similarly summarized abnormalities found at the cellular level. As documented in Table 4 and illustrated in Fig. 10, granular fraction *vf* correlated inversely with clinical index, being much lower in patients with high values of the index. Note as well, from Table 1, that the patients with high clinical index also had high calcium index, which forecasts that the correlation will apply to the cell-level index as well (the prediction could not be tested because calcium indexes were not available for every subject).

The present comparisons involved a small set of patients; they must be extended to larger samples before any consequences can be drawn. The present techniques will allow a rapid extension of the analysis as samples become available. Additional difficult questions can now be addressed; among them: do MHS and other conditions that chronically elevate cytosolic [Ca²⁺] associate with changes in the size, cellular location, or other features of glycogen granules?

Besides glycogen granules, other particles, found by EM, immunofluorescence or other imaging methods could be similarly characterized. Examples of large numbers of protein clusters in confocal microscopic images of developing human myotubes immunostained for RyR1, junctophilin 1, and calpain 1 are in Figs. 2, 3, 6, and 7 of Tammineni et al. (2023). In every scenario envisioned, suitable supervised learning models can replace explicit morphological analysis, provided that the number of images both allows and justifies the initial work of model building and training. In this regard, models must be adapted to the specific tasks and trained specifically for those, through many cycles of refinement. To facilitate these tasks, Data S2 provides all program codes, a detailed protocol, and ancillary computer routines required for pre- and postprocessing. The posting in GitHub includes pretrained weights that can be used as a starting point in a transfer learning approach.

### Online supplemental material

Data S1 includes the description of the structure of categorical classifiers, results obtained with them, their codes, code of utilities for pre- and postprocessing, and a protocol of the complete analysis. Data S2 includes codes of semantic segmenters, a summary of their layers, and utilities for pre- and postprocessing. Data S3 includes the code of the programs assembled for the simulation of granules in a volume and codes for the detection of granules in granule masks and measurement of their diameters.

### Data availability

All original data images used for training and predicting are deposited in Harvard Dataverse as follows: images for training of Locations and their label masks are available at Rios et al. (2024b); images for training of Granules and their label masks are available at Rios et al. (2024c); and all images predicted at Rios and Samsó (2024). A fourth dataset (Rios et al., 2024a) contains all analysis and graphics files (Sigmaplot and Excel) produced for this study. A "data trace" line, appended to every figure and table legend, allows localization and retrieval of corresponding data from the Dataverse. Individual files in the dataset are labeled for traceback to corresponding figures and tables in the manuscript. All these datasets are contained in a dataverse (Rios et al., 2024d) at https://dataverse.harvard.edu/dataverse/JGenPhysiol2024. The annotated code for the four models described is shared in the GitHub repository as Rios, 2023a, 2023b, 2024a, 2023b; b.

## Acknowledgments

Olaf S. Andersen served as editor.

We are grateful to Paul Mielke (Deeplearning.AI) for mentoring us on all things AI, to Amira Klip (Hospital for Sick Kids, Toronto) for teaching us glucose metabolism and advising us at multiple stages of this study, to Clara Franzini-Armstrong (University of Pennsylvania) for long-term teaching and encouragement, and to Biao Zuo (Electron Microscopy Resource Lab, University of Pennsylvania) for expert imaging.

This paper was funded by grants from the National Institute of Arthritis and Musculoskeletal and Skin Diseases (R01 AR 071381 to E. Rios, S. Riazi, and M. Fill [Rush University], R01 AR 072602 to E. Rios, S.L. Hamilton, S.Jung, and F. Horrigan [Baylor College of Medicine], and R01 AR068431 to M. Samsó) and the National Center for Research Resources (S1055024707, to E. Rios and others).

Author contributions: E. Rios: Conceptualization, Data curation, Formal analysis, Funding acquisition, Investigation, Methodology, Project administration, Resources, Software, Supervision, Validation, Visualization, Writing—original draft, Writing—review & editing, M. Samsó: Conceptualization, Data curation, Funding acquisition, Investigation, Resources, Writing—review & editing, L.C. Figueroa: Formal analysis, Investigation, Methodology, Visualization, Writing—review & editing, C. Manno: Data curation, Validation, Visualization, Writing—review & editing, E.R. Tammineni: Data curation, Formal analysis, Investigation, Validation, L.R. Giordano: Formal analysis, Methodology, Visualization, S. Riazi: Conceptualization, Investigation, Methodology, Project administration, Validation, Visualization, Writing—original draft, Writing—review & editing.

Disclosures: The authors declare no competing interests exist.

Submitted: 23 April 2024

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

# Supplemental material

**Provided online are three datasets. Data S1 includes a description of categorical classification models, their Python codes, and utilities for pre- and postprocessing. Data S2 shows Python code of the semantic segmentation modules. Data S3 shows a simulator of granules in a volume and detector and measurer of granules in label masks.**

