## [Peer Review File · The Journal of General Physiology]

Artificial Intelligence approaches to the volumetric quantification of glycogen granules in EM images of human tissue

Eduardo Ríos, Montserrat Samsó, Lourdes Figueroa, Carlo Manno, Eshwar Reddy Tammineni, Lucas Rios Giordano, and Sheila Riazí

Corresponding Author(s): Eduardo Ríos, Rush University

Review Timeline:

Submission Date:	April 23, 2024
Editorial Decision:	May 20, 2024
Revision Received:	May 29, 2024
Editorial Decision:	June 6, 2024
Revision Received:	June 12, 2024
Accepted:	June 13, 2024

Editor: Olaf Andersen

Transaction Report:

DOI: <https://doi.org/10.1085/jgp.202213308>

May 21, 2024

Prof. Eduardo Ríos
Rush University
Physiology & Biophysics
1750 W. Harrison Street
Suite 1279JS
Chicago, Illinois 60605

Re: 202413595

Dear Eduardo,

Your manuscript entitled "Artificial Intelligence approaches to the volumetric quantification of glycogen granules in EM images of human tissue" was seen by two of the reviewers who saw the original version. Both reviewers find that the manuscript satisfactorily addresses the concerns raised in the original reviews and is scientifically acceptable for publication in Journal of General Physiology. Reviewer #1 has no further comments/suggestions. Reviewer #2 identifies areas where the presentation could be clarified; the comments call for a thorough, straightforward revision and formal acceptance will follow when it is modified in accordance with the referees' remarks and our editorial policies.

Please note items that need attention are listed at the bottom of this email (under 'manuscript formatting checklist') and on the attached marked-up pdf file. Please also be sure to include a letter addressing the reviewers' comments point-by-point and a copy of the text with alterations highlighted (boldfaced or underlined). Your manuscript should be a double-spaced MS Word file and include editable tables, if appropriate.

Please submit your final files via this link:
Link Not Available

Thank you for choosing to publish your research in JGP and please feel free to contact me with any questions.

Sincerely,

Olaf S. Andersen, M.D.
On behalf of Journal of General Physiology

Journal of General Physiology's mission is to publish mechanistic and quantitative molecular and cellular physiology of the highest quality; to provide a best in class author experience; and to nurture future generations of independent researchers.

Manuscript formatting checklist:

- MS Word document of text needed (including editable tables)
- MS Word document of supplemental (including figure legends and editable tables)
- Brief Statement describing supplementary information needed (in subsection at end of Materials & Methods)
- Please include a data availability statement preceding the Acknowledgments section. Please see <https://rupress.org/jgp/pages/editorial-policies#data-availability-statement>
- Figures created at sufficient resolution and in acceptable format (including your supplemental Figures). If working in Illustrator, we prefer .ai or .eps file format. If working in Photoshop please use 600dpi/1000dpi .tiff or .psd file format. Minimum resolution at estimated print size: Minimum resolution for all figures is 600 dpi. For figures that contain both photographs and line art or text, 600 dpi is highly recommended. Figures containing only black and white elements (line art, no color, and no gray) should be 1,000 dpi. Maximum figure size is 7 in wide x 9 in high (17.5 x 22.8 cm) at the correct resolution. <https://jgp.rupress.org/fig-vid-guidelines>
- Supplemental figures conforming to same guidelines as manuscript Figures (noted above) and uploaded as individual high resolution images
- If images resemble one from a prior publications, the author must seek permissions (to reproduce or adapt) from the original publisher. [You can resubmit your paper while waiting to hear back from the original publisher but please keep us updated]
- All authors must complete a disclosure form prior to acceptance. A link to complete the form has been sent to all coauthors. Please provide the editorial office with updated email addresses if necessary

Reviewer #1 (Comments to the Authors):

In the revision, the complexity and utility of the technique has been revised and presented in sufficiently detailed manner. My initial questions have been answered satisfactorily.

Reviewer #2 (Comments to the Authors):

The authors have addressed all my concerns satisfactorily. The development of a new semantic segmentation model and the investigation of the influence of particle size on the relationship between pixel fraction and particles per cubic micron, as well as glycogen volume fraction, will provide an important resource. This will improve the research field by enabling objective analyses and facilitating the analysis of numerous images in a cost-effective process. Other researchers will be able to fine-tune and validate the model for other image datasets and purposes. Given the substantial revision and the development of a new semantic segmentation model, I have a few additional minor comments and questions, which are listed below.

1. In the abstract, it is reported that the glycogen concentrations are 11.8 g/l in MHN and 3.9 g/l in MHS. However, in Table 3, the values are 7.0 and 2.3 g/l. Why is there a difference in the reported values? Given the large variability between subjects, the authors should consider reporting the findings in the abstract in a way that aligns with the observations presented in Figures 9 and 10.
2. Table 1: Should ID 277 be listed as MHN?
3. In the paragraph, "The "Locations" segmenter" it could be mentioned that near-SR may also include near-mitochondrial glycogen granules.
4. Page 5: The sentence "In spite of the limited numbers of patients in this sample, the AI analysis found a substantial difference in granular glycogen content of skeletal muscle between the MHS and MHN individuals studied" may need reconsideration. The small sample size could indeed increase the risk of a false positive result and an exaggerated difference. Perhaps the sentence could be revised to reflect this, or it might be best to remove it altogether.
5. Page 5: "The consolidation of Z disks and mitochondria had no incidence in the final assignment of granules to locations, as few granules were found in either region." This finding aligns with the literature, which could be emphasized (Marchand et al). However, how does this statement align with the volume fraction (vf) values in Table 2, where the values for Z-disc and Mitoch are at the same level as the vf values of I-band and near-SR for some subjects? If it is assumed that glycogen granules are not part of the z-disc and mitochondria, could the overall value be adjusted to account for this?
6. Fig 9: Why are some circles red?
7. Given the large variation between the MHS subjects and that this variation may be explained by differences in clinical symptoms, it would be interesting to know the vf of glycogen in the different locations of the two MHS subjects with a clinical index of 6.67. Thus, could the MHS subjects with higher clinical symptoms have a different distribution of glycogen?
8. Fig. 11 and the comparison of the two AI models: The approach of comparing the two models to get insight into the validity of the models are not convincing, because the lack of statistical significance by a t-test may not represent "no major" difference between them. Here it would be more informative with a mean of the differences and a 95% confidence interval for the difference.
9. Discussion: "But, as argued, in EM images small granules cannot be distinguished from horizontally cut "caps" of bigger granules. Thus, the measured diameters bias the actual distributions toward smaller values. This could be a reason for the skew in the measured curve (Fig. 6 D)." Maybe this bias is counterbalanced by the difficulties in detecting small particles. Did the authors validate how well the "granule" model can find small particles? Otherwise, it could be a limitation that in response to glycogen depleting exercise the particles can be reduced in size to much smaller particles than discussed in the manuscript. Marchand et al. found particles as small as 14 nm in diameter.
10. Discussion: Regarding the interesting discussion about the potential of glycogen in other forms the limitations that subsarcolemmal glycogen is not included in the current model and that glycogen was not measured biochemically should be mentioned. However, subsarcolemmal glycogen is reported to only contribute with around 10% of total glycogen, so it is unlikely to explain the difference. But it is relevant information for the reader.

We thank the referees for their kind reading. These are detailed responses to Reviewer #2.

1. In the abstract, it is reported that the glycogen concentrations are 11.8 g/l in MHN and 3.9 g/l in MHS. However, in Table 3, the values are 7.0 and 2.3 g/l. Why is there a difference in the reported values? Given the large variability between subjects, the authors should consider reporting the findings in the abstract in a way that aligns with the observations presented in Figures 9 and 10.

The numbers were wrong in the abstract. Now all numbers have been removed from it.

2. Table 1: Should ID 277 be listed as MHN?

Corrected, thanks.

3. In the paragraph, "The "Locations" segmenter" it could be mentioned that near-SR may also include near-mitochondrial glycogen granules.

We now write “near-SR (S, a region that includes all cytosol in inter-myofibrillar spaces).”

4. Page 5: The sentence "In spite of the limited numbers of patients in this sample, the AI analysis found a substantial difference in granular glycogen content of skeletal muscle between the MHS and MHN individuals studied" may need reconsideration. The small sample size could indeed increase the risk of a false positive result and an exaggerated difference. Perhaps the sentence could be revised to reflect this, or it might be best to remove it altogether.

We considered changing the wording, but we did not do it, as we did point out the errors of both kinds in our statistical measures.

5. Page 5: "The consolidation of Z disks and mitochondria had no incidence in the final assignment of granules to locations, as few granules were found in either region." This finding aligns with the literature, which could be emphasized (Marchand et al). However, how does this statement align with the volume fraction (vf) values in Table 2, where the values for Z-disc and Mitoch are at the same level as the vf values of I-band and near-SR for some subjects? If it is assumed that glycogen granules are not part of the z-disc and mitochondria, could the overall value be adjusted to account for this?

We now mention the agreement of the finding with the published record. We also mentioned that the granules assigned to mitochondria and SR are likely to result from overlap.

6. Fig 9: Why are some circles red?

Circle colors are coded to clinical index; this is now clarified in the figure legend.

7. Given the large variation between the MHS subjects and that this variation may be explained by differences in clinical symptoms, it would be interesting to know the vf of glycogen in the different locations of the two MHS subjects with a clinical index of 6.67. Thus, could the MHS subjects with higher clinical symptoms have a different distribution of glycogen?

The requested distribution is provided with Table 2. There are no anomalies of distribution in the patients of high Clinical Index. To emphasize the observation, the text in Results was revised to “The large difference prevails in every region of the cells and correlates with the severity of the syndrome.”

8. Fig. 11 and the comparison of the two AI models: The approach of comparing the two models to get insight into the validity of the models are not convincing, because the lack of statistical significance by a t-test may not represent "no major" difference between them. Here it would be more informative with a mean of the differences and a 95% confidence interval for the difference.

It is only asserted that “the comparison provided no indication of major errors...”. We believe that this wording is an accurate representation of the implications of the negative result. Note also a full paragraph (starting with “Eqs. 3 and 4...”) in the 2nd page of Discussion, which assesses the implications of the comparison.

9. Discussion: "But, as argued, in EM images small granules cannot be distinguished from horizontally cut "caps" of bigger granules. Thus, the measured diameters bias the actual distributions toward smaller values. This could be a reason for the skew in the measured curve (Fig. 6 D)." Maybe this bias is counterbalanced by the difficulties in detecting small particles. Did the authors validate how well the "granule" model can find small particles? Otherwise, it could be a

limitation that in response to glycogen depleting exercise the particles can be reduced in size to much smaller particles than discussed in the manuscript. Marchand et al. found particles as small as 14 nm in diameter.

The referee may be correct in that missing small granules may offset the bias in our simulation; we did not mention the possibility because we did not find good ways to measure the ability of the model to detect particles of specific size. In any case, the error incurred in with the biased distribution would be minor, as shown by the small difference in $v_f(pf)$ associated with the distributions “Small” and “Large”.

10. Discussion: Regarding the interesting discussion about the potential of glycogen in other forms the limitations that subsarcolemmal glycogen is not included in the current model

We now write “The large excess remains after adding contributions to granular content from subsarcolemmal areas not found in the images analyzed (Marchand et al., 2007; Ørtenblad and Nielsen, 2015).

...and that glycogen was not measured biochemically should be mentioned.

The remarks seem unnecessary as it is clear that the present study does not include biochemical analysis and the comparison is made explicitly with the Burke and Marchand biochemical results.

Thanks again.

June 6, 2024

202413595R1

Dear Eduardo,

Thank you for submitting the revised version of your manuscript, entitled "Artificial Intelligence approaches 1 to the volumetric quantification of glycogen 2 granules in EM images of human tissue" to JGP. The revised manuscript has been seen by me and JGP's Scientific Editor.

Whereas I find that you have addressed most of the reviewers' comments, I also find that you in some cases should be more forthcoming and explicitly acknowledge in the manuscript the uncertainties you acknowledge in your Responses to Reviewers; that is, clarifications included in Responses to Reviewers should be included in the manuscript text because the reviewers are unlikely to be the only people who have these concerns. I found reviewer 1's comments to be insightful, but I am not sure you always paid them justice. Please make sure you do.

Specifically, you should expand your responses to comments 4, 8, 9 and 10.

With respect to the question of two Methods sections, I disagree with you—mostly because the quality of the underlying experimental results will be critical for the outcome of any analysis. You may be correct, that some readers will focus more on the analysis, but I do not see the advantage of separating the Methods section, please merge them.

The Scientific Editor (Nestor Saiz) has two comments that need to be addressed in the manuscript. I paste them in below:

1. The authors claim in the Methods that the code for the analysis has been deposited on GitHub, as I had suggested to them. However, no address is provided for the GitHub repository (here's an example of what it should look like: https://github.com/nestorsaiz/saiz-et-al_2020). Additionally, they mention the data are in a Harvard Dataverse repository, but again, no link to this is provided (or, at least, I don't see it).

2. The authors include a ResearchGate web forum as a reference (line 611). This is not acceptable. I already told them the first time that this cannot be referenced as is. This is a web forum where researchers (or anyone with an account) can post and comment. It's commonly used to request/suggest technical advice or share tips of other kind, but it has the same value as any other Internet forum out there. I asked them to integrate the information they got from that forum into their methods, and include proper references to back it up—something the people posting on ResearchGate don't have to do, and often don't do, but that is expected in peer-reviewed papers like this one. They need to change this, not only this is just bad praxis, but also it doesn't follow our own reference guidelines (<https://rupress.org/jgp/pages/reference-guidelines>)

The requested changes call an additional, but straight-forward, revision. We look forward to receiving the presumably final version.

Please submit your revised manuscript via the link below, along with a point-by-point letter that details your response to the reviewers' and editors' comments, as well as a copy of the text with alterations highlighted (boldfaced or underlined). If the article is eventually accepted, it would include a 'revised date' as well as submitted and accepted dates. If we do not receive the revised manuscript within one year, we will regard the article as having been withdrawn. We would be willing to receive a revision of the manuscript at a later time, but the manuscript will then be treated as a new submission, with a new manuscript number.

Thank you for submitting your interesting research to JGP.

Please submit your revised manuscript, and any associated files, via this link:

Link Not Available

Sincerely,

Olaf

On behalf of Journal of General Physiology

Journal of General Physiology's mission is to publish mechanistic and quantitative molecular and cellular physiology of the highest quality; to provide a best-in-class author experience; and to nurture future generations of independent researchers.